# Structure and Properties of Carboxylated Carbon Nanotubes@Expanded Graphite/Polyethersulfone Composite Bipolar Plates for PEM

**DOI:** 10.3390/nano13142055

**Published:** 2023-07-12

**Authors:** Wenkai Li, Yixin Zhao, Xingchen Pan, Mingqi Liu, Shi Qiu, Zhiyong Xie

**Affiliations:** Carbon-Carbon Composite Materials Research Institute, Powder Metallurgy Research Institute, Central South University, Changsha 410017, China

**Keywords:** carbon nanotubes, carbon nanotube functionalization, graphite/resin composites, bipolar plate, chemical vapor deposition

## Abstract

Composite bipolar plates (BPs) hinder their application in proton exchange membrane fuel cells (PEMFC) because of their poor conductivity and mechanical properties. Nanofillers can effectively solve this problem but often have a limited effect due to their easy agglomeration. In this work, a continuous mesh carboxylated multi-walled carbon nanotube (MWCNT) coating on the surface of graphite was synthesized by chemical vapor deposition (CVD) and carboxylation modification, and the composite BPs were prepared by molding using prepared reticulated carboxylated MWCNTs, expanded graphite, and resin. By optimizing the carboxylation treatment time and the content of the nano-filler, the composite BPs had the best performance at a 15 min carboxylation treatment time and 2.4% filler content. The planar conductivity reached up to 243.52 S/cm, while the flexural strength increased to 61.9 MPa. The thermal conductivity and hydrophobicity were improved compared with the conventional graphite/resin composite BPs, and good corrosion resistance has been demonstrated under the PEMFC operating environment. This work provides a novel nanofiller modification paradigm for PBs.

## 1. Introduction

PEMFC is a new energy battery of interest in the 21st century because of its high efficiency in the conversion of chemical energy into electrical energy directly from hydrogen and oxygen through redox reactions and its environmental friendliness, as the only by-product of this process is water [1,2,3]. However, its practicalization is still slow. At present, PEMFC still has disadvantages such as high cost [4], poor durability [5], and low actual efficiency [6]. To advance the practicalization of PEMFC, it is essential to improve the performance of its components, which mainly consist of the proton exchange membrane, catalytic layer, gas diffusion layer (GDL), and BPs; among them, the bipolar plate occupies most of the mass and manufacturing cost of PEMFC [7,8,9] and assumes the role of conducting electricity between single cells, supporting the structure of this PEMFC, and managing the hydrothermal transport [8,10,11]. Therefore, improving the performance of composite bipolar plates is quite important for improving the overall performance of PEMFC. According to the 2025 DOE requirements, composite BPs should have good planar conductivity (>100 S/cm), high flexural strength (>40 MPa), low area-specific resistance (<0.01 Ω·cm^−2^, and low corrosion current density (<1 μA·cm^2^) under the operating environment of PEMFC (PH = 1) [12].

BPs can be divided into graphite bipolar plates, metal bipolar plates, and composite bipolar plates, according to different materials. A graphite bipolar plate is currently the most mature type of bipolar plate. It has the advantages of high electrical conductivity and corrosion resistance, but it also has problems such as difficult processing and poor mechanical properties [13,14,15,16]. Metal bipolar plates have excellent mechanical properties, processing properties, and electrical conductivity [17,18,19,20,21]. However, because of the poor corrosion resistance of metal BPs compared to graphite BPs, methods such as corrosion-resistant coatings are needed to overcome the degradation of fuel cell durability due to corrosion resistance [7,22,23,24]. Composite BPs are usually made using graphite and various resins such as polyimide, epoxy, and other polymers, and have good processability and corrosion resistance [13,25,26,27]. At the same time, composite BPs have better processability than graphite BPs, which can be prepared easily and inexpensively by molding or injection molding [28,29]. However, it is difficult for composite BPs to have both good electrical conductivity and bending strength, so how to improve the performance of composite plates is still a problem that needs to be solved.

In order to improve the performance of composite BPs, the most common method is to adjust the ratio of raw materials to meet the needs of users. H.S. Lee et al. studied the effect of the proportion of graphite particles on the electrical properties of composite BPs [30]. S.R. Dhakate studied the effect of phenolic resin powder on the performance of composite BPs under different volume fractions [31]. The mechanical properties and electrical conductivity of the two cannot be achieved at the same time, therefore, the main two types of methods to solve the problem include overall structure optimization and the addition of conductive filler. Through the optimization and rational design of the internal structure of composite BPs, it is possible to have both good electrical conductivity and bending strength, such as the composite BPs with a sandwich structure prepared by Mao et al. using Expanded graphite (EG)/Ni@Melamine foam (MF)/EG with electrical conductivity (320 S/cm) and bending strength (56 MPa) [9]. Low filler and highly conductive composite BPs with synergistic segregated structures were prepared by Hu et al. By controlling the segregated structure, the electrical conductivity (177.87 S/cm) and flexural strength (49.16 MPa) were prepared to meet the 2025 DOE standard [32]. The addition of new conductive fillers can also improve the conductive properties of composite BPs without compromising flexural strength. A variety of fillers have been proven to be effective, such as conductive carbon black [33], carbon fibers [34], carbon nanotubes [35], graphene [36], etc. The preparation and performance of polypropylene/multi-walled carbon nanotubes/carbon nanofiber composite BPs were studied by C.A. Ramírez-Herrera et al., and their performance was able to meet DOE requirements [37]. Hu et al. also investigated the performance of polyvinylidene fluoride/graphite composite BPs with the addition of MWCNTs, and the results showed that the addition of MWCNTs could effectively enhance the performance of the composite BPs [38].

Although nano-fillers can effectively improve the performance of composite BPs, the improvement of the performance of composite BPs is limited due to their easy agglomeration and the fact that they are not easily wetted by resin [33,39]. In order to solve the problems of easy agglomeration and poor wettability of nanofillers and fully improve the performance of composite BPs, well-dispersed mesh MWCNTs were prepared by CVD in this study to solve the problem that nanofillers are not easy to disperse. The MWCNTs were made to take on carboxyl groups by acid treatment to effectively improve their wettability with resin. By improving these two problems of nanofillers, the electrical conductivity and toughening properties of nanofillers can be fully exploited, and the function of enhancing electrical conductivity and strength can be fully utilized. Therefore, the overall performance of composite BPs can be improved.

## 2. Materials and Methods

### 2.1. Materials

Expanded graphite (EG) was purchased from Shenzhen Hanhui Graphite Co., Ltd. (Shenzhen, China) and used as a substrate for CVD and as a raw material for the preparation of composite BPs. The cobalt nitrate (analytical pure reagents (AR)), nitric acid (guaranteed reagents (GR)), and sulfuric acid (guaranteed reagents (GR)) were purchased from Aladdin Reagent Co. (Shanghai, China) and the polyethersulfone (PES) resin was purchased from Guangdong Huiteng New Material Co., and was used for preparing BPs.

### 2.2. Preparation of a Network of MWCNTs Wrapped with an EG Surface

The flow of the whole experiment is shown in Figure 1. Expanded graphite was weighted and put in a beaker. Cobalt nitrate of 5 wt% by mass of graphite was added to the solution. Excess alcohol was also added, and the solution was sonicated to allow sufficient dissolution of cobalt nitrate. The solution was then heated and stirred in a water bath at 70 °C until the solution was paste-like, and the solution was dried in an oven at 50 °C to obtain expanded graphite with cobalt nitrate particles loaded on the surface.

The dried powder was placed in a quartz boat and put into a tube furnace for chemical vapor deposition. The deposition process was divided into two stages of temperature rise: firstly, the temperature was increased to 400 °C, and the gas was fed according to the argon: hydrogen flow rate ratio of 2:1 to reduce the cobalt nitrate on the graphite surface to monolithic cobalt. The hydrogen was turned off, and the argon gas was fed. Then, the temperature was slowly warmed up to 800 °C, pass argon, acetylene, and hydrogen in the ratio of 300:30:10 sccm, and dropped to room temperature under argon protection. The yield was also calculated based on the weight difference before and after deposition and was used for subsequent experiments.

### 2.3. Preparation of Carboxylated Carbon Nanotubes Coated with Graphite

Graphite containing 2 g of MWCNTs was weighed, and 10 mL of concentrated nitric acid was added. Then, 30 mL of concentrated sulfuric acid was slowly added to the mixed solutions for the carboxylation treatments. After the solution was fully mixed, it was placed in a water bath at 75 °C and heated for carboxylation at different times. The labeled samples are shown in Table 1. MWCNTs with different carboxylation treatment times were labeled as C_x_-MWCNTs. x is the treatment time. While investigating the effect of treatment time on the composite BPs, composite BPs were prepared using C_x_-MWCNTs with a mass fraction of 1% with 40% EG and 60% PES, and the related properties were investigated.

### 2.4. Preparation of Composite BPs

The mass fraction of MWCNTs was adjusted by mixing 40 wt% of PES resin with 60 wt% of EG and by controlling the amount of untreated EG and treated EG. The mixed powder was placed in a 40 mm × 40 mm mold, and 2 g of powder was poured into the mold to produce the composite BP. In order to investigate the effect of the content of carbon nanotubes on the performance of composite BPs, different contents of carboxylated carbon nanotubes were used to prepare composite BPs, and the samples were labeled as CM-BPs, while as a control group, pure MWCNT powder was purchased and directly added to the powder by mixing to prepare composite BPs for the control group, and the samples were labeled as PM-BPs. Based on the conclusions of Section 3.1, C_15min_-MWCNTs were used for the experiments in this section. The details of all samples numbered by grouping are shown in Table 2. Composite BPs prepared using only graphite and resin without any nanofiller were named N-BPs for studying the effect of nanofiller on composite BPs. The content of carboxylated MWCNTs was controlled by controlling the ratio of deposited graphite (i.e., MWCNTs + EG) to pure EG.

### 2.5. Characterization

The PHI-5000versaprobe III X-ray photoelectron spectrometer was used for qualitative and quantitative analysis of the effect of carboxylation treatment time on MWCNTs. An Evolution Raman spectrometer and Ultima IV X-ray diffractometer were used to study the effect of carboxylation treatment time on the microstructure of MWCNTs, such as crystallinity and graphitization. The Raman spectroscopy test range is from 100 to 4000°, and the XRD test range is from 5 to 90°. Double-beam scanning electron microscopy (SEM, JSM-7600F) was used to characterize the microscopic morphology of graphite and composite BPs. Spherical aberration transmission electron microscopy (TEM, JEOL ARM-200) was used to characterize the nanoscale morphology of MWCNTs and the integrity of their lattices.

To study the conductivity of composite BPs, the ST2258C four-probe resistivity tester was used for testing, as shown in Figure 2A. To study the mechanical properties of composite BPs, the Instron 3369 Universal Mechanical Tester was used to test the flexural strength. The test was performed by the three-point bending method, and the samples were made into long strips of 40 mm × 10 mm × 0.5 mm.

The interface contact resistance (ICR) with GDL and area-specific resistance (ASR) measurement adopts the test standards in the DOE standard, and the FT361SJB tester was used for testing ASR and ICR. The BPs are sandwiched between two pieces of carbon paper, and, subsequently, the carbon paper is sandwiched between two copper plates. The voltage is measured after passing a constant current to characterize the contact resistance between the composite BPs and the carbon paper (i.e., GDL).

The test principle and test circuit are shown in Figure 2B, and the contact resistance between the composite BP and the GDL is calculated according to the following equation:(1)$$R=\frac{V\times A_{s}}{I^{2}}$$
(2)$$R_{total}=2R_{GDL}+2R_{GDL}{}_{/Cu}+2R_{GDL/b}+R_{b}$$
(3)$$R_{sys}=2R_{GDL}+2R_{GDL}{}_{/Cu}+2R_{GDL/GDL}$$
(4)$$ICR\;{=R}_{GDL/b}=\left(R_{\mathrm{total}}-R_{\mathrm{sys}}-R_{b}\right)$$
(5)$$ASR=2ICR+R_{b}$$

The contact resistance is calculated by Equations (1)–(5). The overall resistance is calculated by Equation (1) based on the current density, and the overall error resistance ***R_sys_*** is tested by a blank experiment (i.e., without putting in BPs), and the ***ICR*** (i.e., ***R_GDL/b_***) is obtained by subtracting the overall resistance ***R_total_*** from the error resistance ***R_sys_*** and the known resistance of the composite BPs, and the ***ASR*** is obtained by adding the overall resistance of the composite BPs.

To study the surface hydrophobicity of BPs, contact angle tests were performed on composite BPs. The JC2000A contact angle tester was used to test the contact angle of BPs. The thermal conductivity was then tested using the ISO 22007-2 test standard using the HOT DISK TPS 2500S, and the composite BPs were prepared as specimens with a thickness of 2 mm for testing.

The compaction density of the composite BPs was calculated by weighing the mass and measuring the volume, while its full porosity was estimated using the BET. The Mike ASAP2460 Automatic Specific Surface and Porosity Analyzer BET was used to analyze the pore size distribution, porosity, average pore size, and other parameters of the composite BP. The samples were prepared in small blocks of 1 mm in length, width, and height for testing and were degassed for 8 h at 120 °C.

Corrosion testing of composite BPs was performed on a CHI660E electrochemical workstation with three electrodes. The electrolyte was 0.5 M H_2_SO_4_. The constant potential polarization and dynamic potential polarization methods were used to evaluate the corrosion resistance of the composite BPs, respectively. The constant potential polarization method was performed at −0.1 V and 0.6 V in order to evaluate the corrosion resistance of their anodes and cathodes, respectively. The kinetic potential polarization method, on the other hand, was performed in the range of −0.5 V to 0.9 V with a scan rate of 2 mV/s.

## 3. Results and Discussion

### 3.1. Effect of Carboxylation Time on the Performance of Layered MWCNTs and Composite BPs

Unlike the granular MWCNTs commonly used as additives, the MWCNTs prepared by the CVD method naturally demonstrate a good dispersion and a large specific surface area, so the reaction rate was very fast during the carboxylation process and the results were very sensitive to time. To investigate the carboxylation time for MWCNTs, the morphology of MWCNTs coated with graphite under different time treatments was studied, and the results are shown in Figure 3.

It is found from Figure 3 that MWCNTs coated the graphite surface. With the increase in carboxylation time, the coating layer was gradually destroyed, and its fibrous structure was also gradually destroyed by acid. The fibers also gradually changed from long nanotubes, as Figure 3A–C shows, that formed a continuous network structure, to short, dispersed nanotubes, as Figure 3D–F shows. This is because the CVD-prepared MWCNTs covered the surface of the substrate-expanded graphite in a thin layer [40], and thus were more dispersed, had a larger contact area with the acid, and had a faster reaction rate compared to the agglomerated MWCNTs. In contrast, too long a reaction time led to the shearing of MWCNTs and even the destruction of their structure.

Combined with the TEM results, it is found that when the treatment time was short, as shown in Figure 3G, the MWCNTs still had complete lattice stripes, while the lattice stripes gradually became discontinuous as the carboxylation time gradually rose, as shown in Figure 3H,I. When the treatment time reached 25 min, the lattice stripe of the MWCNTs was no longer complete, but was disconnected, and there was a large amount of amorphous carbon on the surface of the MWCNTs. This indicates that the crystal structure of MWCNTs will be gradually destroyed with the increase in treatment time.

To investigate the effect of a strong acid treatment on the surface of MWCNTs, the carboxylated MWCNTs at different times were characterized by FT-IR spectroscopy, Raman, XPS, and XRD, and the results are shown in Figure 4.

From the FT-IR of Figure 4A, it is found that the sample shows obvious absorption peaks at 1720 cm^−1^ and 3300 cm^−1^ after the carboxylation treatment, and the absorption peak at 1720 cm^−1^ has a typical **C=O** stretching vibration absorption peak, while the hydroxyl stretching vibration peak is typical at 3300 cm^−1^, which indicates that the sample had taken on **C=O** and **-OH** groups after the treatment.

The crystal integrity and graphitization of MWCNTs can be measured by the ratio of the integrated area of the D peak (1380 cm^−1^) to the G peak (1520 cm^−1^) of Raman spectroscopy. The smaller the area ratio of the D peak to the G peak, the higher the graphitization and the closer the crystal structure is to that of ideal graphite. The results of the Raman spectra are shown in Figure 4B, and the calculated results of the integral area ratio are shown in Figure 4E. From the results, it is found that the graphitization degree gradually decreased with the increase in carboxylation treatment time, which was due to the destruction of the graphite crystal structure by the strong oxidizing acid. The treatment time from 0 to 15 min had little effect on the graphitization degree, and the peak area ratios at 5, 10, and 15 min were 0.116, 0.157, and 0.176, respectively. After the treatment time of more than 20 min, the ratio of D to G peak area increases sharply, which indicates that the crystal structure of MWCNTs was severely damaged beyond this treatment time. The decrease in graphitization was very detrimental to electrical and thermal conductivity. The XRD results were similar to the Raman spectra, and the characteristic summits of the graphite structure gradually disappeared with the increase in treatment time. When the treatment time reached 25 min, the characteristic peaks of the crystalline surface **(004)** had completely disappeared, and new characteristic peaks were generated on the plots [41], as Figure 4C shows, which indicates that the destruction of the crystal structure of MWCNTs was very serious at this time.

The XPS profiles and the elemental contents calculated from the peak areas are shown in Figure 4D,E. The oxygen contents for 5, 10, 15, 20, and 25 min were 15.47, 18.76, 20.78, 28.14, and 28.59 (atomic %). This indicates that the oxygen content of the MWCNT cladding layer was increasing with the increase in carboxylation treatment time. The charge correction, followed by fitting the C_1s_ peak of the sample by split peak fitting, revealed that the sample has a characteristic **O-C=O** chemical bond peak at 288.5. Combined with the results of FT-IR spectra, it is known that MWCNTs had carboxyl groups attached, and the percentage of carboxyl groups increased with the increase in treatment time.

A too long treatment time led to the destruction of the MWCNTs cladding layer and reduced the graphitization of the MWCNTs, which reduced the planar conductivity of the composite BPs and increased the contact resistance. In contrast, shorter treatment times lead to an incomplete reaction and a limited effect on the improvement of wettability. To investigate the effect of treatment time on the properties of composite BPs for further experiments, 1 wt% of MWCNTs as additives were mixed with EG and PES resins to prepare composite BPs, and the conductivity and mechanical properties were tested.

The results of Figure 4F show that with the increase in treatment time, the conductivity first decreased slowly and then decreased sharply after 15 min, while the mechanical properties first increased and then decreased. The decrease in electrical conductivity was because the electrical conductivity of carboxylated MWCNTs was inferior to that of pure MWCNTs when no agglomeration occurred and all were in dispersed form. However, the bonding between carboxylated MWCNTs and resin was better because the surface had carboxyl groups with good affinity to resin [42,43], so the mechanical properties increased within a certain range. However, with the extension of treatment time, the structure of MWCNTs was destroyed and became irregular, short-cut fiber-like, or even indeterminate flocculent, so the effect of improving mechanical properties decreased and even had the opposite effect.

From the results, MWCNTs treated for 15 min effectively increased the flexural strength (~19.5%) with less decrease in conductivity (~5.6%), and therefore MWCNTs prepared at this time were used in the subsequent experiments.

### 3.2. Effect of Carboxylated Carbon Tube Cladding on the Performance of BPs and Comparison with Pure MWNCTs

In order to study the dispersion of carboxylated carbon nanotubes in the composite BPs and the effect on the actual performance of the composite BPs, the composite BPs were prepared with different contents of encapsulated carboxylated carbon tubes, and the performance test results are shown in Figure 5.

Figure 5A demonstrates the effect of carboxylated MWCNT content versus pure MWCNT content on the planar conductivity of composite BPs. When the MWCNT content was low, the pure MWCNTs were not agglomerated, and, according to the previous results, the pure MWCNTs had better intrinsic conductivity, so the conductivity of PM-BPs with low MWCNT content was better than that of CM-BPs. When the MWCNT content was raised above a certain value, the MWCNTs could not be effectively dispersed during the preparation of composite BPs because the excess MWCNTs could not be wetted by the resin [44], and the loose clusters were easily formed, as in Figure 6D,E. In contrast, the prepared carboxylated MWCNTs were dispersed in a reticular pattern on the graphite surface, as Figure 6A,B shows, facilitating electron transport and filling the gap [45] between the particles. More importantly, the carboxyl groups on the surface improved the interfacial compatibility between the PES and EG [46,47,48], so that agglomeration occurs at higher MWCNT contents, as Figure 6H shows, and the improvement of electrical conductivity was also more obvious. Overall, for CM-BPs, the conductivity improved to 243.52 ± 2.88 S/cm at 2.4%, which is 2.79% lower than 250.53 ± 3.05 S/cm for PM-BPs at 1.6% and 11.18% higher than 217.69 S/cm for N-BPs.

Benefiting from the improved compatibility and toughening effect of MWCNTs, the mechanical properties of CM-BPs were more improved than those of PM-BPs. The mechanical properties of CM-BPs were improved from 48.58 MPa to 61.93 ± 1.22 MPa for PM-BPs, as Figure 5B shows. In agreement with the results reported in many works [49,50,51,52], the enhancement of mechanical properties of pure MWCNTs for composite BPs was smaller compared to planar conductivity, and at higher contents, it even caused degradation of mechanical properties due to severe agglomeration effects. The prepared dispersed network of carboxylated MWCNTs on the surface of EG not only facilitated the stress tolerance and pore filling, but also effectively bound to the two phases as a third phase filled between PES and EG, thus effectively improving the flexural strength.

The ASR test results were similar to the planar conductivity, and both sets of samples at the appropriate MWCNT contents can meet the 2025 DOE requirements (7.09 Ω·cm^2^ of G_60_R_40_CM_2.4_, 8.04 Ω·cm^2^ of G_60_R_40_PM_2.4_), as Figure 5C shows. At low MWCNT contents, the ASR was effectively reduced because both MWCNTs and reticulated carboxylated MWCNTs could play the role of electron transport. As the MWCNT content increased, the surface of the PM-BPs was destroyed by severe agglomeration, and a large number of voids were generated by the inability to be wetted [53,54], as shown in Figure 6F, which increased the ICR between the GDL and therefore raised the ASR. In the case of CM-BPs, although the surface was not cracked due to wettability as shown in Figure 6C,I, the high contents of carboxylated MWCNTs on the surface reduced the flatness of the surface and increased the ICR, and the high contents of carboxylated MWCNTs also led to a decrease in intrinsic conductivity. The combination of both factors led to an increase in ASR.

Since MWCNTs are inherently better thermally conductive materials [55,56,57], the addition of MWCNTs will have a positive effect on the overall thermal conductivity of the material. MWCNTs can act as fillers to fill pores and reduce thermal resistance, and are also excellent thermal conductivity additives themselves. From the results of Figure 5D, the effect of agglomeration on thermal conductivity was less in the experimental range (0–4%), and the thermal conductivity continues to increase with the increase of MWCNTs.

Hydrophobicity is important for the water management function of composite BPs in PEMs [58,59], and the contact angle with water is usually used to assess hydrophobicity. The results of the hydrophobicity test are shown in Figure 5E. For PM-BPs, the hydrophobicity was improved at lower MWCNT contents compared to N-BPs due to the inherent better hydrophobicity of MWCNTs as well as the filling effect [60,61]. At higher MWCNT content, as mentioned before, the resin layer on the surface of the composite BPs was destroyed and a large number of cracks were generated, as Figure 6I shows. Therefore, the hydrophobicity will keep decreasing. In contrast, for CM-BPs, although the surface was not destroyed due to good wettability with PES, a large amount of hydrophilic carboxylated MWCNTs with EG existed on the surface at higher MWCNT contents, as Figure 6G shows, so the hydrophobicity rapidly decreased as the MWCNT content rose.

In order to deeply investigate the mechanism of MWCNTs for the internal filling of composite BPs and the pore rise caused by wettability, whole-pore BET tests were conducted on the samples, and the whole-pore distribution of composite BPs was obtained by DFT calculations. From the results of Figure 5F, the pores of the PM-BPs were more than those of the CM-BPs at almost all diameters at the proportion of MWCNTs that show agglomeration (4%), because the carboxylated MWCNTs were more compatible with PES [62,63] and the agglomeration effect was less. At a lower percentage of MWCNTs (2.4%), there were fewer (>60 nm) pores at CM-BPs, while PM-BPs had slightly fewer (<60) nm small pores. This is due to the fact that the better wettable carboxylated MWCNTs were more dispersed in the sample and had a better filling effect on the pores [64,65,66] so the larger pores were more filled. The reduction in porosity was beneficial for mechanical properties and electron transport.

In conclusion, as shown in Table 3, the addition of carboxylated MWCNTs resulted in a more pronounced improvement in performance than the direct addition of pure MWCNTs, and a higher content of nanofillers was required for the emergence of agglomerates. A cross-sectional comparison of similar studies also shows some progress in their performance. By optimizing the carboxylation treatment time and the content of the nano-filler, the composite BPs had the best performance at a 15 min carboxylation treatment time and 2.4% filler content. The planar conductivity reached up to 243.52 S/cm, while the flexural strength increased to 61.9 MPa.

### 3.3. Effect of Carboxylated MWCNTs on the Corrosion Resistance of BP Performance and Comparison with Pure MWNCT

The chemical stability of the composite BPs is closely related to the durability of PEMFC, so the composite BPs need to have good corrosion resistance under acidic conditions [22,67,68]. In order to investigate the corrosion resistance of composite BPs under the PEMFC operating environment, constant potential polarization tests and dynamic potential polarization tests were carried out. The results are shown in Figure 7.

From the results of the constant potential polarization at +0.6 V and −0.1 V, as shown in Figure 7A,B, the polarization currents of both G_60_R_40_CM_2.4_ and G_60_R_40_PM_2.4_ were low after a period of sample polarization compared with the currents of N-BPs without any MWCNTs added, and the currents of both G_60_R_40_CM_4_ and G_60_R_40_PM_4_ with a large number of MWCNTs added were higher. Both reticulated carboxylated MWCNTs and pure MWCNTs have good stability in weakly acidic environments and can fill defects and pores and protect the surface of composite BPs [69], so the corrosion resistance of composite BPs improved in moderate amounts. While the higher content of MWCNTs and carboxylated MWCNTs will lead to the destruction of the surface of BPs because of the influence of agglomeration and the reaction active site, resulting in an increase in corrosion current density.

The test results of the dynamic potential polarization were similar to the constant potential, and the addition of the appropriate amount of MWCNTs and carboxylated MWCNTs both favor the decrease of corrosion current density. Because the agglomeration of carboxylated MWCNTs was more difficult to occur, the corrosion current density of G_60_R_40_CM_2.4_ was smaller (0.602 μA/cm^2^) with the same content of MWCNTs, but both groups can meet the DOE requirements. In contrast, the corrosion current density decreased in the group with the addition of excess MWCNTs and carboxylated MWCNTs compared to the control group N-BPs, as Figure 7C,D shows, indicating that both excess agglomerated MWCNTs and carboxylated MWCNTs led to a decrease in the stability of BPs in the simulated PEMFC environment.

## 4. Conclusions

In this study, graphite coated with reticular MWCNTs was constructed by the CVD method, and reticular carboxylated MWCNTs were prepared by controlling the carboxylation time. The prepared carboxylated MWCNTs were used to prepare composite BPs, and unlike the conventional composite BPs with direct addition of MWCNTs, the carboxylated MWCNTs were well dispersed between the graphite and the resin, and the wettability with the resin was significantly improved. The mechanical properties, electrical conductivity, thermal conductivity, and corrosion resistance of the composite BPs were improved by the conductivity and ductility enhancement of the carboxylated MWCNTs. In addition, the results of the corrosion resistance and hydrophobicity tests showed that the hydrophobicity was improved and the corrosion resistance was in accordance with DOE standards. In conclusion, this work provides a new method and structure to effectively improve the dispersion of nanofillers in composite BPs and the wettability with resin components and to prepare high-performance resin/graphite composite BPs for PEMFC.

## Figures and Tables

**Figure 1 nanomaterials-13-02055-f001:**
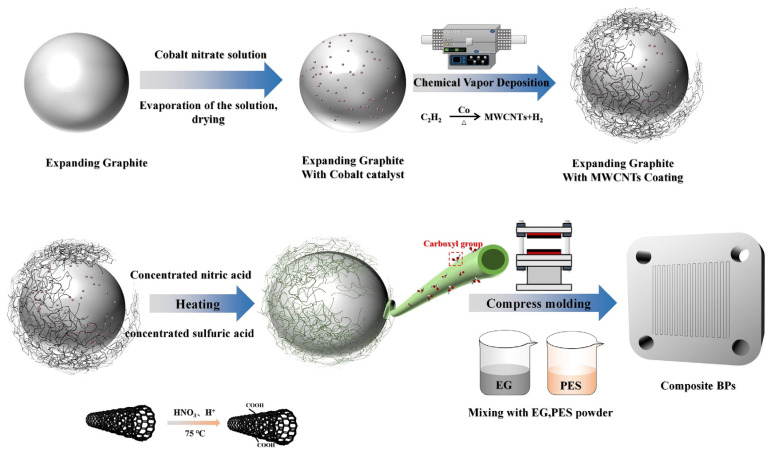
Preparation of carboxylated carbon nanotube cladding layers and composite BPs.

**Figure 2 nanomaterials-13-02055-f002:**
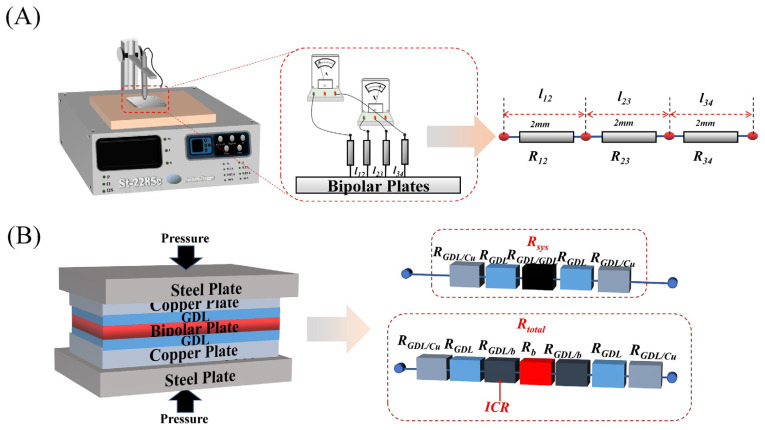
Performance tests of composite BPs: (**A**) planar conductivity test; and (**B**) ASR test.

**Figure 3 nanomaterials-13-02055-f003:**
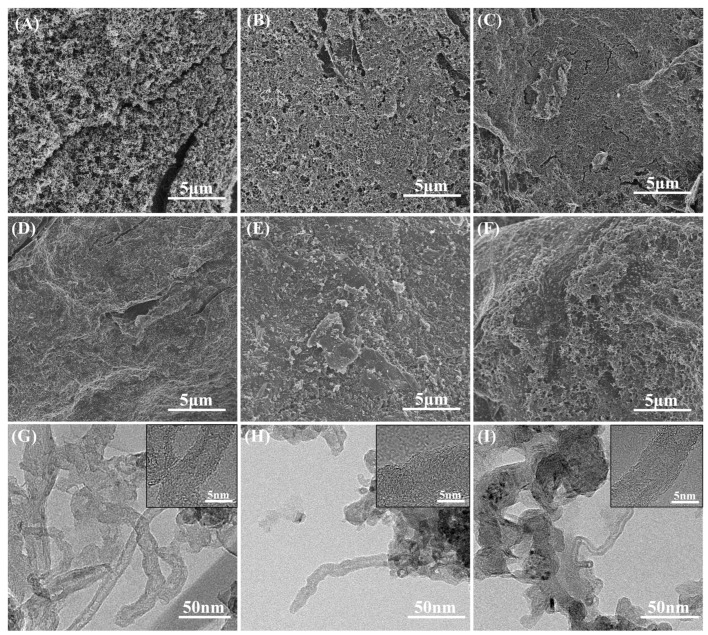
MWCNT cladding layers after different time carboxylation treatments: (**A**) 0 min; (**B**) 5 min; (**C**) 10 min; (**D**) 15 min; (**E**) 20 min; and (**F**) 25 min. TEM images of different time carboxylation treatments: (**G**) 5 min; (**H**) 15 min; and (**I**) 25 min.

**Figure 4 nanomaterials-13-02055-f004:**
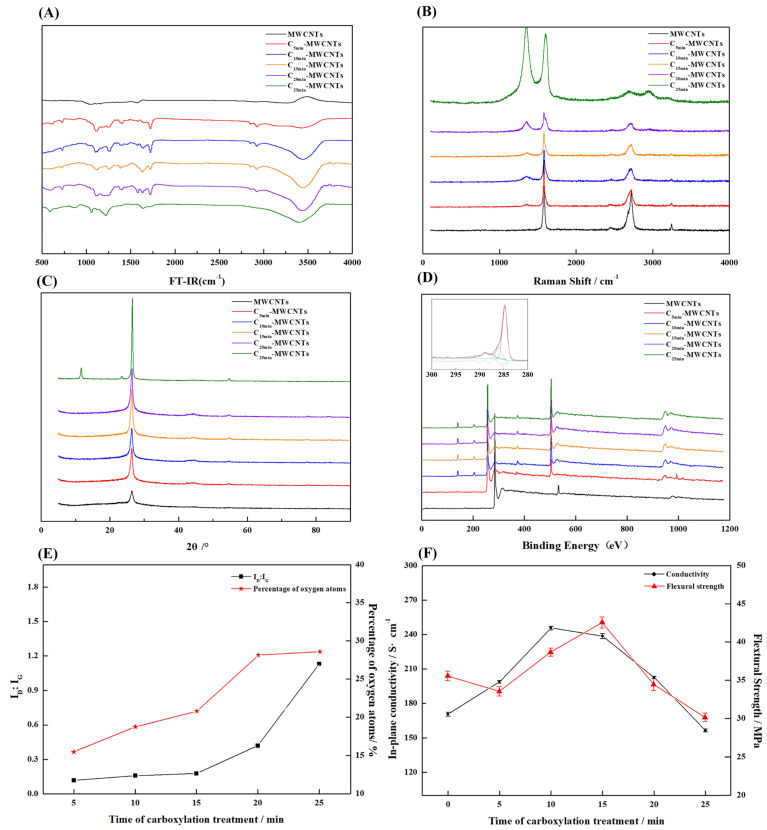
MWCNTs treated with different times of carboxylation: (**A**) FT-IR; (**B**) Raman spectra; (**C**) XRD; (**D**) XPS, as well as the peak fitting results; (**E**) Raman peak area integration ratio and elemental oxygen content (atomic %); and (**F**) conductivity and flexural strength of composite BPs prepared at different times of carboxylation.

**Figure 5 nanomaterials-13-02055-f005:**
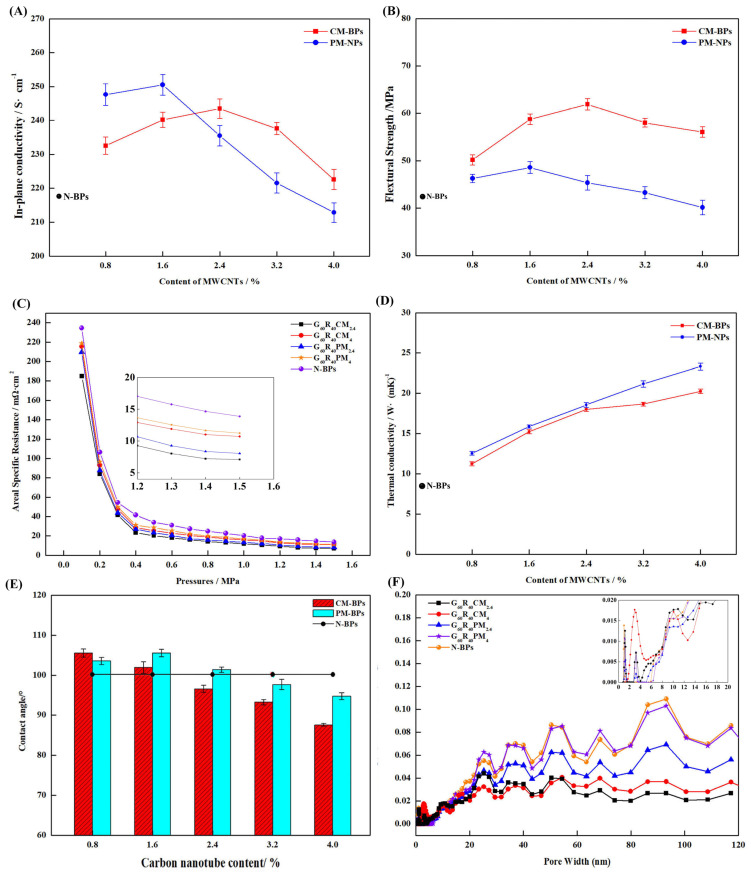
Effect of carboxylation-treated MWCNTs and pure MWCNT content on the properties of composite BPs: (**A**) planar conductivity; (**B**) flexural strength; (**C**) ASR; (**D**) thermal conductivity; and (**E**) hydrophobicity. (**F**) The volume fraction of micropores and mesopores.

**Figure 6 nanomaterials-13-02055-f006:**
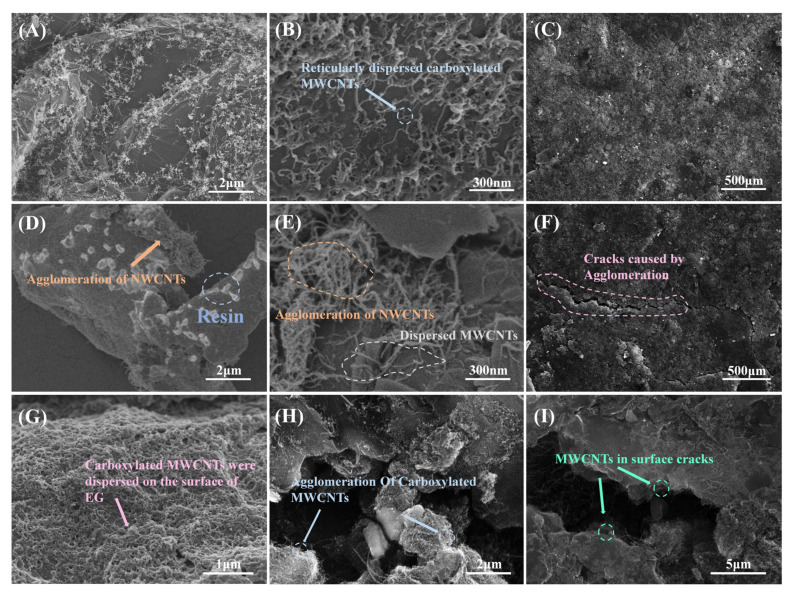
Structure of composite BPs: (**A**,**B**) the interior of CM-BPs; (**C**) the surface of CM-BPs; (**D**,**E**) the interior of PM-BPs; and (**F**) the surface of PM-BPs. (**G**) Carboxylated carbon nanotubes distributed on the surface of graphite. (**H**) Agglomeration of carboxylated MWCNTs at high content. (**I**) Surface cracks caused by agglomeration of MWCNTs.

**Figure 7 nanomaterials-13-02055-f007:**
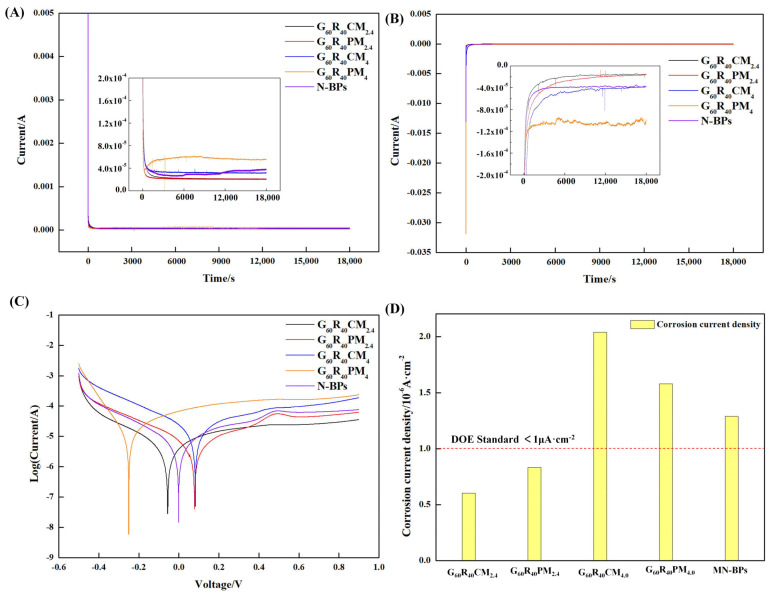
Electrochemical properties of composite BPs: (**A**) constant potential polarization curve at +0.6 V; (**B**) constant potential polarization curve at −0.1 V; (**C**) Tafel curve; and (**D**) corrosion current density.

**Table 1 nanomaterials-13-02055-t001:** MWCNTs with different carboxylation treatments.

Materials	Carboxylation Treatment with H_2_SO_4_ and HNO_3_ Time (min)
MWCNTs	0
C_5min_-MWCNTs	5
C_10min_-MWCNTs	10
C_15min_-MWCNTs	15
C_20min_-MWCNTs	20
C_25min_-MWCNTs	25

**Table 2 nanomaterials-13-02055-t002:** Compound BPs with different additives.

Groups	Sample	EG wt%	PES wt%	Carboxylated MWCNTs wt% of EG + PES	Pure MWCNTs wt% of EG + PES
N-BPs	-	60%	40%	-	-
CM-BPs	G_60_R_40_CM_0.8_	60%	40%	0.8%	0
	G_60_R_40_CM_1.6_	60%	40%	1.6%	0
	G_60_R_40_CM_2.4_	60%	40%	2.4%	0
	G_60_R_40_CM_3.2_	60%	40%	3.2%	0
	G_60_R_40_CM_4_	60%	40%	4%	0
PM-BPs	G_60_R_40_PM_0.8_	60%	40%	0	0.8%
	G_60_R_40_PM_1.6_	60%	40%	0	1.6%
	G_60_R_40_PM_2.4_	60%	40%	0	2.4%
	G_60_R_40_PM_3.2_	60%	40%	0	3.2%
	G_60_R_40_PM_4_	60%	40%	0	4%

**Table 3 nanomaterials-13-02055-t003:** Principle performance comparison of composite BPs.

Sample	Electrical Conductivity /S·cm^−1^	Flexural Strength /MPa	Materials
N-BPs	217.87	42.34-	PES, expanded graphite, MWCNTs
G_60_R_40_CM_0.8_	232.57	50.17	
G_60_R_40_CM_1.6_	240.19	58.76	
G_60_R_40_CM_2.4_	243.52	61.9	
G_60_R_40_CM_3.2_	237.62	58.03	
G_60_R_40_CM_4_	222.57	56.08	
G_60_R_40_PM_0.8_	247.65	46.27	
G_60_R_40_PM_1.6_	250.53	48.58	
G_60_R_40_PM_2.4_	235.52	45.35	
G_60_R_40_PM_3.2_	221.56	43.27	
G_60_R_40_PM_4_	212.86	40.13	
EG/Ni@MF/EG-EP [9]	320	56	Expanded graphite, (EG)/Ni@Melamine foam (MF)
G34CB6 [32]	177.87	49.16	Polyvinylidene fluoride, conductive carbon black, graphite
G82.5CB2 [33]	25.7	34.8	Polyethylene, graphite, carbon black

## Data Availability

The data presented in this study are available on request from the corresponding author.

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
