# Peer review of "Structure and Properties of Carboxylated Carbon Nanotubes@Expanded Graphite/Polyethersulfone Composite Bipolar Plates for PEM"

_nanomaterials, 2023, doi:10.3390/nano13142055_

Round 1
Reviewer 1 Report
The manuscript deals with the synthesis of graphite coated with reticular MWCNTs for the enhancement of the conductivity of composite bipolar plates. The topic is of practical interest. The investigation carried out is of high quality. The experimental design is logical and described in details. Conclusions are fulli supported with the experimental data obtained.
There are several remarks to be taken into account prior to the acceptance of the manuscript to publication.
1. Abbreviations need comprehensive revision.
- First of all, the abbreviation should be inserted at first mention both in the abstract and in the text of the manuscript.
- Both abbreviated and full descriptions are existed now in the text decreasing its readability.
- PES abbreviation should be explained at first mention.
- Abbreviations used one-two times should be removed.
- Abbreviations for synthesized materials should be the same throughout the manuscript including figures and tables.
2. Section 2.1, the abbreviated purity of the reagents (AR and GR) is better to show as "AR purity" and "GR purity".
3. Figures 4, 5, and 7 quality is insufficient. The inserted parts and legends are unreadable.
4. Figure 4E and 7D, each point needs error bar.
5. Lines 306-308, the conductivity values have to be shows as the average value±SD. Similar changes are needed for the mechanical properties at the line 311.
6. Lines 417-418, lines 436-445, the text should be removed as there are no Supplementary Materials, Appendix A, and Appendix B in the manuscript.
English is acceptable. Minor revision is enough.
Author Response
Thank you for your review and we have made the following revisions based on your valuable comments:
Point 1: Abbreviations need comprehensive revision
Response: We have corrected the full form of abbreviations that appear for the first time in the abstract and body of the text, and removed abbreviations that are used only 1-2 times. We have also fixed the problem of inconsistent abbreviations in the charts and text tables.
Point 2: In section 2.1, the abbreviated purity of the reagents (AR and GR) is better shown as "AR purity" and "GR purity".
Response: We have revised these issues to the form (GR purity), (AR purity)
Point 3: Figures 4, 5, and 7 quality is insufficient. The inserted parts and legends are unreadable.
Response: We have re-uploaded larger images and increased the resolution from 300dpi to 600dpi.
Point 4: In Figures 4E and 7D, each point needs an error bar.
Response: The experimental conclusion in Figure 4-e was calculated from the basic ratio of XPS and Raman peak area, while the corrosion current density in Figure 7-dwas obtained by tangential fitting of the Tafel curve, so there may not be error bar due to multiple experimental measurements.
Point 5: In lines 306-308, the conductivity values have to be shown as the average value±SD. Similar changes are needed for the mechanical properties at the line 311.
Response: This has been revised to the form of value±SD.
Point 6: In lines 417-418, and lines 436-445, the text should be removed as there are no Supplementary Materials, Appendix A, and Appendix B in the manuscript.
Response: These texts have been correctly deleted.
Once again, we would like to express our sincere appreciation for your review of our work.
Reviewer 2 Report
The manuscript presents experimental investigation devoted to development of material for bipolar plate of the fuel cells. The proposed strategy of the material production includes usage of expanded graphite as a template substrate for multiwall carbon nanotube synthesis.
Obtained composite material is charaterized in the wrok to obtain its structural and elctrical characteristics. Also electrochemical perfomance of obtained compoistes is investigated.
All together presented results seems interesting and instructive for readers dealing with the problems related to the fuel cells and carbon-based composites.
Description presented in the manuscript is quite detailed and conclsuions are quite justified.
So, the manuscript may be published as it is.
The manuscript presents experimental investigation devoted to development of material for bipolar plate of the fuel cells. The proposed strategy of the material production includes usage of expanded graphite as a template substrate for multiwall carbon nanotube synthesis.
Obtained composite material is charaterized in the wrok to obtain its structural and elctrical characteristics. Also electrochemical perfomance of obtained compoistes is investigated.
All together presented results seems interesting and instructive for readers dealing with the problems related to the fuel cells and carbon-based composites.
Description presented in the manuscript is quite detailed and conclsuions are quite justified.
So, the manuscript may be published as it is.
Author Response
Thank you for your review and recognition of our work.
Reviewer 3 Report
Please, see file attached.

Author Response
Thank you for your review and recognition of our work. Based on your comments, we have made the following revisions.
Point 1: The abstract should be completely rewritten and contain the way BPs are prepared, as well as the preparation parameters when BPs achieve the best performance.
Response 1: We have completely reworked the abstract and added the filler content for optimal performance and the method of preparation of BPs based on the comments.
Point 2: The background of the research on BPs and PEMFC in the introduction is incomplete.
Response 2: We have revised the relevant issues in accordance with your comments, as follows:
- In lines 31 to 33, we added a brief introduction of how PEMFC works, and an explanation of its environmental friendliness.
- In lines 39-41, we added a brief description of the specific roles that BPs assume in PEMFC.
- In lines 43 to 45, we add a detailed elaboration of the technological goals that need to be achieved by 2025 for of BPs
- Lines 55-57, a description of the types of materials commonly used for composite BPs has been added.
- In 57,58 lines, a description of the common preparation methods for composite BPs was added.
- In lines 88-95, we added a description of the objectives of this study, the problems to be solved, and the innovative points compared to the traditional methods of improving composite properties by direct addition of nanofillers.
- Based on the previous comments, we have also added some references to the literature.
Point 3: The method and experiments of the research on BPs and PEMFC in the introduction are incomplete.
Response 3: We have revised the relevant issues in accordance with your comments, as follows:
- In lines 99-103, we have revised the problem of unclear presentation of the relevant material, and (AR) and (GR) have been revised to (AR purity) as well as (GR purity).
- We have revised the product from “layer-wise mesh MWCNTs” to the more accurate description of the “Network of MWCNTs wrapped with EG surface”.
- The repeated paragraph created by our oversight has been removed.
- In lines 126 to 128, and Table 1, the relevant expressions regarding the carboxylation treatment have been rewritten.
- Table 2, the expression for the content of MWCNTs in the sample has been rewritten and is now changed from the total mass fraction to the mass fraction of EG+PES.
Point 4: What is the GDL in the ICR characterization?
Response 4: The GDL in the ICR and ASR tests was a gas diffusion layer. Due to our unclear explanation and improper omission, the test conditions are unclear, and we have added relevant explanations in the text, including the material of the GDL itself.
Thank you again for your review of our work, which has helped us to correct many errors and misrepresentations.
Round 2
Reviewer 3 Report
Please, see my comments attached

Author Response
Thank you for your review and recognition of our work. Based on your comments, we have made the following revisions. We apologize for not being able to answer all of your previous questions due to an oversight on our part.
Point 1: Reagent description unclear.
Response 1: In lines 99-103, we have revised the problem of unclear presentation of the relevant material, and (AR) and (GR) have been revised to (Analytical Rure Reagents (AR)) as well as (Guaranteed Reagents (GR)). The company of the reagent was also indicated.
Point 2: Which MWCNTs with carboxylation treatments has been used in Table 2?
Response 2: The carboxylation time for the carboxylated carbon nanotubes used in Table 2 was determined based on the experimental findings in Section 3.1 for a 15 min carboxylation treatment time. We have revised the description in the article and added the relevant description where Table 2 is located.
Point 3: What is the meaning of G60R40CM0.8 in Table 2? 60 wt. % of Graphite, 40 wt. % of Resin, and 0.8 wt. % of Graphite coated with carboxyl MWCNT?
Response 3: Since the carboxylated MWCNTs were encapsulated on EG, we obtained the net content of MWCNTs by weighing the sample weight gain before and after CVD in order to precisely control the content of MWCNTs, and by controlling the proportional ratio between the deposition product (i.e., EG + MWCNTs) and pure EG to make the content of MWCNTs an exact 0.8 wt%. We have added the relevant expressions in the text.
Point 4: The contact angle test was used to assess the hydrophobicity of the samples and the JC2000A contact angle tester was used to test the contact angle of BPs?
Response 4: We have revised the presentation to make it clearer. Since the surface of BPs is often in contact with water in the operating environment of PEMFC, the hydrophobicity of composite BPs is often characterized by measuring the contact angle between the surface of BPs and water.
Point 5: Plots of Figure 4 and 5 are difficult to see because the size.
Response 5: We reformatted Figures 4 and 5 and re-uploaded the images in higher quality (from 300dpi to 900dpi).
Point 6: What is the composition of N-BPs in Figure 5?
Response 6: N-BPs are blank controls prepared without adding any nano-fillers, using only pure expanded graphite and PES, for studying the effect of adding nano-fillers on the composite BPs among themselves. We describe their relevant constituents in Section 4.2 and Table 2.
Point 7: A Table with the principal properties achieved with the best sample must be presented and compared with the state of art of similar composite BPs.
Response 7: We have added a new table (Table 3) at the end of Section 3.2 and added principal performance comparisons with the control group in this paper, as well as comparisons with similar work by other investigators.
Again, we apologize for our oversight and thank you for your careful review, which is very helpful in improving the quality of our work.
Round 3
Reviewer 3 Report
About point 5)
From Figure 5A, 5B, 5D (5D I think is not well labeled) readers deduce that the composition of N-BP is 0.8 %
Author Response
Thank you for your review and recognition of our work. Based on your comments, we have made the following revisions.
Point 1: From Figure 5A, 5B, 5D (5D I think is not well labeled) readers deduce that the composition of N-BP is 0.8 %
Response: Figure 5 has been redrawn to make it less likely to be misinterpreted, and in addition, the chart serial numbers have been relabeled.
Thank you very much for your patience and careful review.
